

# An inexpensive, customizable microscopy system for the automated quantification and characterization of multiple adherent cell types

Vishwaratn Asthana[1], Yuqi Tang[1], Adam Ferguson[1], Pallavi Bugga[1], Anantratn Asthana[2], Emily R. Evans[1], Allen L. Chen[1], Brett S. Stern[1] and Rebekah A. Drezek[1]

[1] Department of Bioengineering, Rice University, Houston, TX, USA
[2] Department of Molecular & Cell Biology, University of California, Berkeley, Berkeley, CA, USA

Corresponding author
Vishwaratn Asthana,
vishwaratn.asthana@rice.edu

## ABSTRACT

Cell quantification assays are essential components of most biological and clinical labs. However, many currently available quantification assays, including flow cytometry and commercial cell counting systems, suffer from unique drawbacks that limit their overall efficacy. In order to address the shortcomings of traditional quantification assays, we have designed a robust, low-cost, automated microscopy-based cytometer that quantifies individual cells in a multiwell plate using tools readily available in most labs. Plating and subsequent quantification of various dilution series using the automated microscopy-based cytometer demonstrates the single-cell sensitivity, near-perfect $R^2$ accuracy, and greater than 5-log dynamic range of our system. Further, the microscopy-based cytometer is capable of obtaining absolute counts of multiple cell types in one well as part of a co-culture setup. To demonstrate this ability, we recreated an experiment that assesses the tumoricidal properties of primed macrophages on co-cultured tumor cells as a proof-of-principle test. The results of the experiment reveal that primed macrophages display enhanced cytotoxicity toward tumor cells while simultaneously losing the ability to proliferate, an example of a dynamic interplay between two cell populations that our microscopy-based cytometer is successfully able to elucidate.

## INTRODUCTION

Cell quantification assays are essential components of most biological labs, and are used for a variety of applications, including cytotoxicity, viability, and proliferative studies. Though these assays have improved significantly since the advent of hemocytometers, they still suffer from a number of apparent limitations. Current cell quantification assays can be divided into two major classes: metabolic and cell counting. Metabolic assays, though originally designed to assess cell viability, are often used to indirectly assess cell number. Metabolic assays like MTT (3-(4,5-dimethylthiazol-2-yl)-2,5-diphenyltetrazolium bromide) or alamarBlue, both of which use the reductive

environment of the cell to convert dye reagents into detectable colored products, are relatively easy to perform, and can provide additional information on cell health that a counting assay may not. However, they are not ideal for certain experimental setups, as they have a limited dynamic range and are prone to confounding interference in the presence of certain chemicals (*Chakrabarti et al., 2000*; *Doak et al., 2009*; *Hamid et al., 2004*; *Ulukaya, Colakogullari & Wood, 2004*; *Vistica et al., 1991*). These assays also do not always align well with the DNA content of the cell—a parameter that correlates strongly with cell number—limiting the cell quantification potential of these assays (*Quent et al., 2010*). Cell counting assays on the other hand, though generally more manually intensive, are more representative of actual cell counts than metabolic-based proxy assays (*Chan et al., 2013*). Flow cytometry, often considered the gold standard for cell counting and analysis, is an especially powerful technique for quantifying individual cells, and is one of the few modalities capable of identifying cell population counts in both a mono-culture and co-culture setup (*Gedye et al., 2014*; *Gerashchenko, 2008*; *Gerashchenko & Howell, 2013*).

Co-culture systems are fundamental to studying any kind of cell-to-cell interaction. Many areas of research could benefit immensely from co-culture setups—including biomaterials, immunology, and cancer biology—if better characterization and quantification methods were available for these studies (*Bidarra et al., 2011*; *Miki et al., 2012*). Most approaches to co-culture are restrictive, and often require that plated cells be physically separated via a transwell insert or microfluidic chamber that only permits the exchange of media (*Arrigoni et al., 2016*; *Goers, Freemont & Polizzi, 2014*; *Katt et al., 2016*). Yet physical contact has been shown to be important for studying the interactions of many cell types in a variety of physiological contexts (*Cruickshank et al., 2004*; *de Goer de Herve et al., 2010*; *Gerashchenko & Howell, 2003*; *Holt, Chamberlain & Grainger, 2010*; *Suzuki et al., 2004*). Most cell populations do not behave independently and a better understanding of the interaction between multiple cell types in a system will help improve our understanding of many physiological phenomena.

Most studies that utilize a true co-culture setup with physical contact rely on flow cytometry to quantify individual cell types. Unfortunately, flow cytometry has several drawbacks that apply not only to co-culture setups but mono-culture setups as well. As a starting point, flow cytometers are fairly sophisticated; as a result, these instruments are generally expensive and often require skilled upkeep (*Nasi et al., 2015*). Flow cytometry also requires cells to be in suspension; thus, the majority of experiments that are conducted on adherent cells in multiwell plates require trypsin treatment for cell detachment. Trypsin, however, can damage cells and cleave extracellular markers that may be used for cellular identification or other forms of analysis (*Gedye et al., 2014*). Additionally, certain cell types are not amenable to trypsin treatment and as a result require manual cell scraping, a process that is prone to human error. Cell scraping can also mechanically damage some cell types, leading to erroneous results with certain assays that are used in combination with flow cytometry: for example, false positives with membrane permeable cell viability assays such as propidium iodide (*Batista et al., 2010*; *Bundscherer et al., 2013*). Furthermore, cells in suspension have the potential to stick

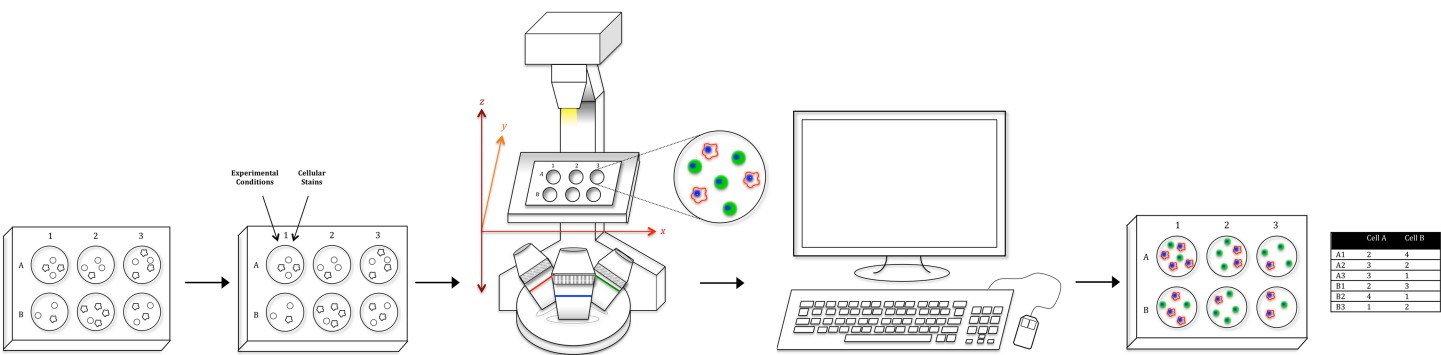

**Figure 1  Workflow schematic.** Using tools available in most labs, we have created a relatively simple optical counting setup for the quantification of multiple adherent cell types in a multiwell plate. After plating cells, adding condition, and fluorescent staining, the proposed system uses a standard fluorescent microscope to capture whole-well images. Image data is then run through ImageJ for pre-processing and CellProfiler for analysis to generate both absolute cell counts for every well as well as morphological values. Furthermore, utilizing a combination of staining techniques, multiple cell types in a co-culture setup (Cell A & B as an example) can be uniquely identified and absolutely counted.

to each other forming aggregates that can interfere with analysis. All together, the need to bring adherent cells into suspension makes flow cytometry less than ideal for many types of studies.

An additional limitation specific to co-culture setups is that typical flow cytometry obtains relative counts rather than absolute counts, i.e., each cell type is expressed as a percent of the total sample assayed. This may generate misleading results when comparing counts for multiple cell types between conditions.

To address the shortcomings of the various cell quantification assays, we have put together a relatively simple optical counting setup with tools readily available in most labs (Fig. 1). The proposed system uses a standard fluorescent microscope to quantify individual cells on a multiwell plate with superior sensitivity, resolution, and dynamic range. The system utilizes established staining techniques to label cells, and subsequently quantifies every cell in the entire experimental space, including the well edge, via whole-well imaging, thus precluding the need for trypsinization or cell scraping. The image data is then run through ImageJ for preprocessing and then analyzed in CellProfiler, a free-to-use cell segmentation software, to generate absolute cell counts for every well. In addition, by utilizing a combination of staining techniques, multiple cell types (even those with complex morphologies) can be uniquely identified and absolutely counted, permitting the setup of more complicated co-culture experiments that were not previously feasible, one of the strongest aspects of the proposed system (*Krtolica et al., 2002*; *Spink et al., 2006*).

It should be noted that other optical counting systems have been developed to address the aforementioned limitations of flow cytometry. However, the optical system we present here still bears a number of advantages over these pre-sold cytometric platforms. First, for labs that already possess a fluorescent microscope, the microscopy-based cytometer is a relatively small investment to the overhead cost of buying a prebuilt optical counting system or flow cytometer. Further, unlike most commercial counting systems,

the microscopy-based cytometer discussed here is flexible in the assays and cell types that can potentially be analyzed, and highly customizable in both setup and analysis, permitting the extraction of more relevant information per experiment. In this way, biological workflow is not limited by the vendor-specific restrictions of pre-built systems, but rather expanded to include all functionalities of standard fluorescent microscopes.

Many image cytometers currently on the market also exhibit a smaller dynamic range and reduced sensitivity compared to the system we present here. Lastly, most commercial cytometric platforms are not able to discriminate multiple distinct cell populations, with potentially complex morphologies, within a single well. This is largely due to the inability of most commercial cytometers to accurately segment non-spherical morphologies, thus restricting their applicability for most co-culture setups. A more in-depth comparison of our microscopy-based cytometer to commercially available systems can be found in the Discussion section as well as in Text S1.

## MATERIALS AND METHODS

### Cell culture and plating

JC CRL 2116 mouse adenocarcinoma cells were obtained from ATCC (American Type Culture Collection, Manassas, VA, USA) and maintained in Dulbecco's modified Eagle medium supplemented with 10% fetal bovine serum (FBS) and 1% penicillin/streptomycin. J774.A1 mouse macrophages were obtained from ATCC and maintained in Roswell Park Memorial Institute Medium (RPMI), supplemented with 10% FBS and 1% penicillin/streptomycin. Both cells were grown at 37 °C in 5% $CO_2$.

For plating, cells were trypsinized (JC CRL 2216 cells) or scraped (J774.A1 cells) from their flasks and quantified manually using a bright-line hemocytometer (Sigma-Aldrich, St. Louis, MO, USA). Due to the variability present in counts obtained using a hemocytometer, established values were primarily used to determine the approximate concentration of the primary stock from which precise subsequent dilutions were performed. Dilution series were generated by pulling cells from the previous stock and diluting in fresh media. Triplicates of each dilution were then plated on either a 48- or 12-well Corning Costar flat bottom cell culture plate (Thermo Fisher Scientific, Waltham, MA, USA) and given 24 h to attach to the plate surface.

### Co-culture experiments

J774.A1 cells were seeded at $2.5 \times 10^4$ cells/well in RPMI media in a 48-well plate. Immediately following seeding, cells were exposed either to 1 μg/mL of lipopolysaccharide (LPS) (Sigma-Aldrich, St. Louis, MO, USA), 0.1 μg/mL of mouse interferon gamma (IFNγ) (BioLegend, San Diego, CA, USA), both 1 μg/mL of LPS and 0.1 μg/mL of IFNγ, or neither. After a 24-h incubation, the wells were washed twice with phosphate buffered saline. JC CRL 2116 cells were then added at $1 \times 10^4$ cells/well in RPMI media to every well. Cells were processed and imaged after 24 h.

## Cell Staining

For assays requiring only a nuclear stain, cell processing was performed immediately prior to imaging. After firm cell adhesion, media was removed from the wells by inverting the plate and cells were fixed for 15 min in BD Cytofix (BD Biosciences, San Jose, CA, USA). Fixative was removed by washing the plate twice with Hank's balanced salt solution (HBSS). Cells were then stained for 5 min in a 2.5 μg/mL 4′,6-diamidino-2-phenylindole (DAPI) stain solution (Thermo Fisher Scientific, Waltham, MA, USA). The plate was again washed twice with HBSS and finally resuspended in HBSS for imaging.

Experiments involving multiple stains, including nuclear, cytoplasmic, and surface/antibody stains, required a slightly different protocol. Using the immune co-culture experiment as an example, JC CRL 2116 cells were stained in their cell culture flask with 20 μM of Vybrant carboxyfluorescein diacetate succinimidyl ester (Vybrant CFDA SE) (Invitrogen, Carlsbad, CA, USA) the day prior to plating using the manufacturer-recommended protocol. After completion of the experiment and immediately prior to imaging, media was removed from the wells by inverting the plate, and the plate was subsequently blocked with a 1% bovine serum albumin (BSA) (Sigma-Aldrich, St. Louis, MO, USA) solution in HBSS for 15 min. HBSS supplemented with calcium was found to help prevent cell detachment prior to cell fixation. Cells were then incubated with a 5 μg/mL phycoerythrin (PE) anti-mouse CD11b antibody (BioLegend, San Diego, CA, USA) diluted in 1% BSA HBSS for 1 h to stain the J774.A1 cells. After incubation, the antibody solution was removed by inverting the plate after which the wells were washed twice with 1% BSA HBSS. The cells were then fixed for 20 min in BD Cytofix. Fixative was removed by washing the plate twice with HBSS. Cells were then stained for 5 min in a 2.5 μg/mL DAPI stain solution. The plate was again washed twice with HBSS and finally resuspended in HBSS for imaging.

## Acquiring whole-well images

All imaging was performed on a Nikon Eclipse Ti–E inverted fluorescent microscope with motorized $x$, $y$, and $z$ stage (Nikon Instruments Inc., Melville, NY, USA). Images were captured using a NAMC 10× objective and Andor Zyla 4.2 sCMOS camera (Andor Technology, Belfast, Northern Ireland). DAPI/Hoechst (excitation: 360/40, emission: 460/50, dichroic mirror: 400), Green Fluorescent Protein (GFP) (Ex: 470/40, Em: 525/50, DM: 495), and Texas Red (Ex: 560/55, Em: 645/75, DM: 595) filter cubes (Nikon Instruments Inc., Melville, NY, USA) were used to image cells stained with DAPI, Vybrant CFDA SE, and PE anti-mouse CD11b antibody, respectively. Using the associated Nikon software, NIS-Elements, an automated macro was set up for whole-well acquisition. First, an $x$–$y$ coordinate list demarcating the center of every well was generated by manually determining the center of the first and last well of the plate and dividing these values by the number of rows and columns. The coordinate list will vary based on the type of multiwell plate but only needs to be generated once. Using a 10× objective and 1,600 × 1,600 pixel region of interest, 10 × 10 images for a 48-well plate and 19 × 19 images for a 12-well plate were tiled together to create a whole-well image. The center of each well serves as the origin point of the tiled images as well as the autofocus point.

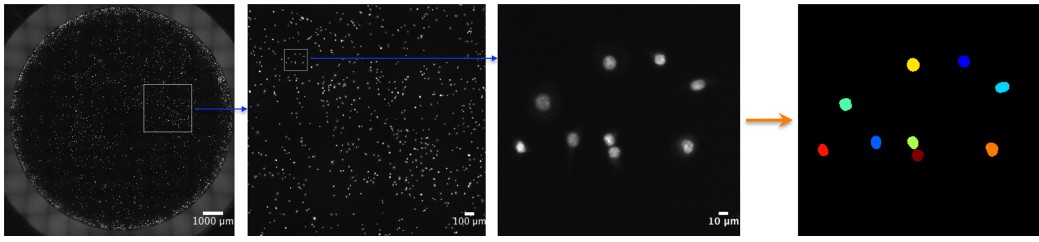

**Figure 2 Automated, whole-well imaging.** Using a fluorescent microscope with motorized *x*, *y*, and *z* stage, images spanning the entire well can be sequentially captured and stitched together to produce a high resolution, whole-well image. In the image series above, JC CRL 2116 cell nuclei stained with DAPI were imaged on a 48-well plate with a 10× objective and stitched together. Sequential zooms of the whole-well image demonstrate the high resolution of the photo, which enables single-cell counting and analysis using a cell segmentation software like CellProfiler (used to generate the final mask).

## Processing whole-well images

Images were preprocessed using ImageJ (National Institute of Health, Bethesda, MD, USA). The ring of autofluorescence around the well edges was removed using the *Subtract Background* function with a rolling ball radius of 50 pixels. Fluorescent channels that were too faint for analysis were occasionally made brighter using the *Enhance Contrast* function. If images were stitched during image acquisition, they were subsequently cropped into 100 smaller images using the *Montage to Stack* function at which point they were transferred to CellProfiler (Broad Institute, Cambridge, MA, USA), specifically CellProfiler version 2.2.0, for segmentation. ImageJ macros (Macro S1) and CellProfiler codes (CellProfiler Code S1 and CellProfiler Code S2) are provided as supplementary files. A more detailed explanation of relevant ImageJ and CellProfiler functions can be found in the supplementary section (Text S1).

## Statistics

All experiments were conducted in triplicates. Error bars represent the standard deviation of replicate experiments. Linear regressions and associated coefficients of determination ($R^2$) were derived using Excel. For the JC CRL 2116 and J774.A1 co-culture experiment, Turkey's HSD test (JMP Pro 13) was used to calculate statistical significance (set at $\alpha = 0.05$) between conditions.

## RESULTS

### Accuracy of nuclear quantification in conjunction with whole-well imaging

Many studies take representative or random images that represent only a small portion of the entire well to make a claim about differences in cell count/viability between conditions. Instead of manually acquiring a series of random or representative images, which is both tedious and potentially inaccurate, it is possible—using a fluorescent microscope with a motorized *x*, *y*, and *z* stage—to automate capture of the whole well using movement in the *x*- and *y*-plane to image the well and movement in the *z*-plane to autofocus (Fig. 2).

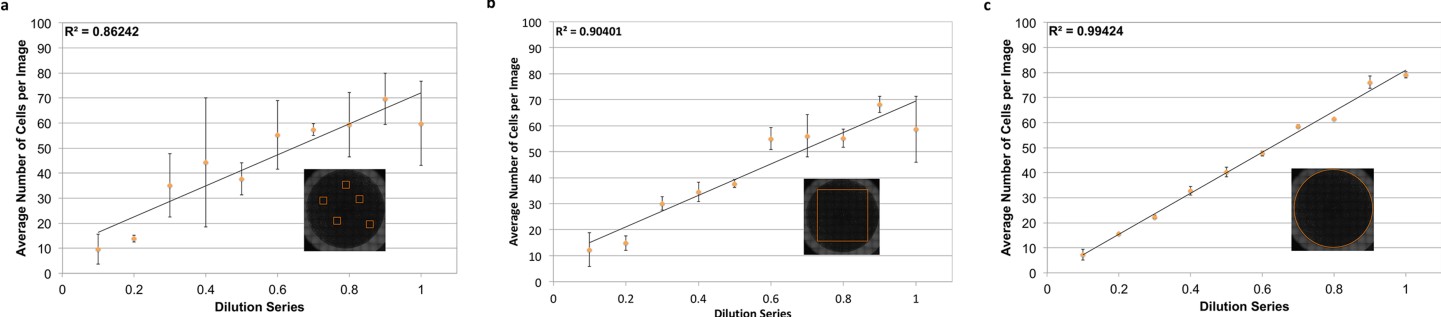

**Figure 3 Power of whole-well imaging and nuclear quantification.** A linear dilution series of JC CRL 2116 cells going from 10,000 cells/well down to 1,000 cells/well was plated on a 48-well plate and given 24 h to adhere to the surface. Whole-well images of cell nuclei stained with DAPI were then captured and processed to produce either (A) five random images, (B) box crop, or (C) whole-well images, which were then used to determine cell count and generate a linear curve. Accuracy improves significantly as the percent of the experimental space being assayed increases. Error bars represent the standard deviation between triplicate conditions.

To count individual cells, wells were stained with a DAPI nuclear stain. Nuclei are often roundly shaped and spaced from adjacent nuclei by the cell cytoplasm and membrane, making segmentation relatively straightforward. In addition, there is usually one nucleus per cell, making nuclear segmentation ideal for cell counting. To demonstrate the power of nuclear counting and whole-well imaging, a linear dilution series of JC CRL 2116 mouse adenocarcinoma cells going from 10,000 cells/well all the way down to 1,000 cells/well was plated; an experiment spanning one order of magnitude. A total of three linear curves were then generated from five random images, a box crop of the whole-well image, or the whole-well image itself, and used to determine cell count (Fig. 3). Accuracy improves significantly as the percentage of the experimental space being assayed increases, with whole-well imaging displaying a near-perfect $R^2$ and minimal error between replicates.

## Establishing dynamic range and sensitivity

The microscopy-based cytometer obtains accurate counts by counting individual cells. In theory it should be able to count anywhere from one cell to the confluency limit of the well plate the experiment is conducted in with single-cell precision. To demonstrate the dynamic range of the system, a dilution series of JC CRL 2116 cells going from 100,000 cells/well all the way down to 100 cells/well were plated in a 12-well plate; an experiment spanning three orders of magnitude (Fig. 4). The system performs strongly within the confluency limit of the plate with a dynamic range limited only by the surface area of the multiwell.

To validate the sensitivity of the system, a linear dilution series starting at 100 cells/well and going down to 1 cell/well was also plated. However it was found that at such low concentrations, it was not feasible to reliably plate the desired number of cells. Accordingly, the cells that were plated were manually counted in the brightfield channel and compared with CellProfiler counts obtained from segmenting nuclei in the DAPI channel (Fig. 5). The system demonstrates single-cell sensitivity and resolution with most deviations attributable to human error when manually counting.

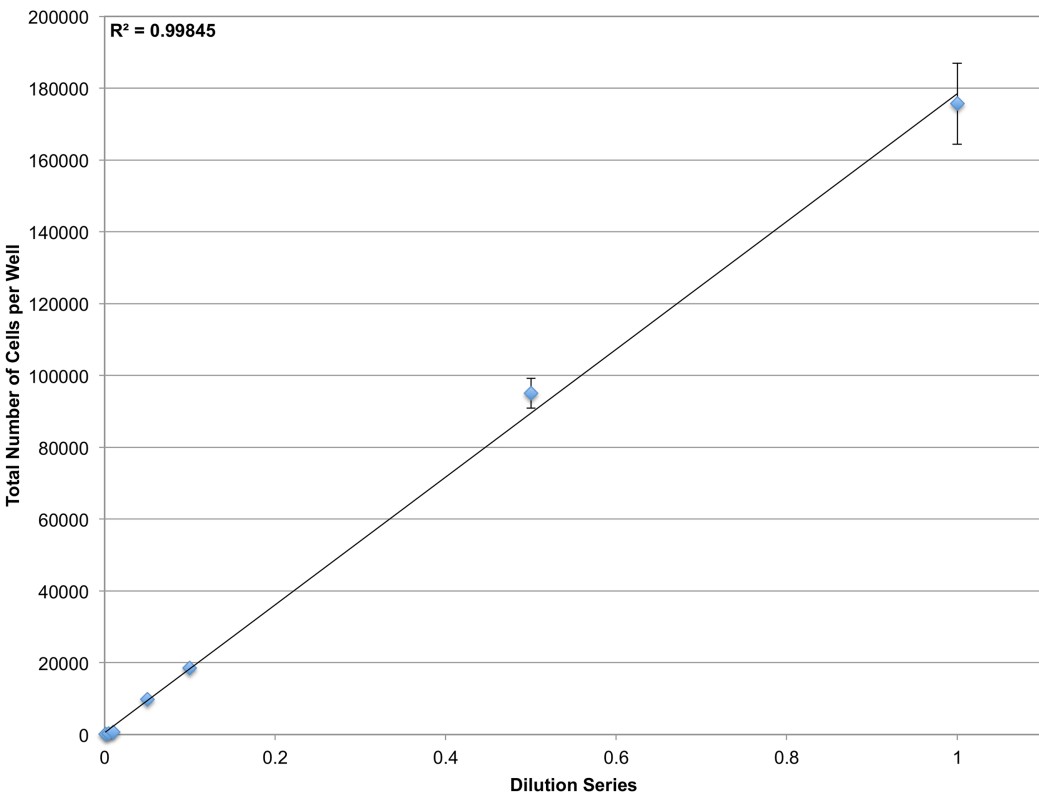

**Figure 4 Determining dynamic range.** A dilution series of JC CRL 2116 cells going from 100,000 cells/well down to 100 cells/well were plated on a 12-well plate (an experiment spanning three orders of magnitude) and given 24 h to adhere to the surface. Cell nuclei stained with DAPI were then quantified and used to generate a linear curve. The system performs robustly over a wide dynamic range and is limited only by the surface area of the multiwell. Error bars represent the standard deviation between triplicate conditions.

## Counting using surface and cytoplasmic stains

In order to count multiple cell types in a co-culture experiment, additional cell stains are required. Cell stains can be grouped into one of three categories: nuclear, surface, and cytoplasmic. Because most nuclear stains non-specifically stain DNA and compromise cell viability, they cannot be used to differentiate cell types. Instead, vital cytoplasmic dyes like Vybrant CFDA SE, or antigen-specific surface stains, such as fluorophore-conjugated antibodies, need to be used.

Surface and cytoplasmic staining alone can be used to differentiate cells, but primary segmentation of these stains is fairly difficult. Factors including inhomogeneous staining, cell contact with neighboring objects, and complex cell morphologies make segmentation less than ideal. One way to overcome this limitation is to use easily identifiable and spatially resolvable nuclei delineated in one fluorescent channel as a seed/primary object to guide detection of the cell border/secondary object outlined in a separate fluorescent channel (*Jones, Carpenter & Golland, 2005*; *Vincent & Soille, 1991*). However, when plating more than one cell type, for example in a co-culture setup, this approach alone is insufficient because every nuclei will generate a secondary object regardless of whether there is an associated cell in CellProfiler (*Carpenter et al., 2006*). To overcome this

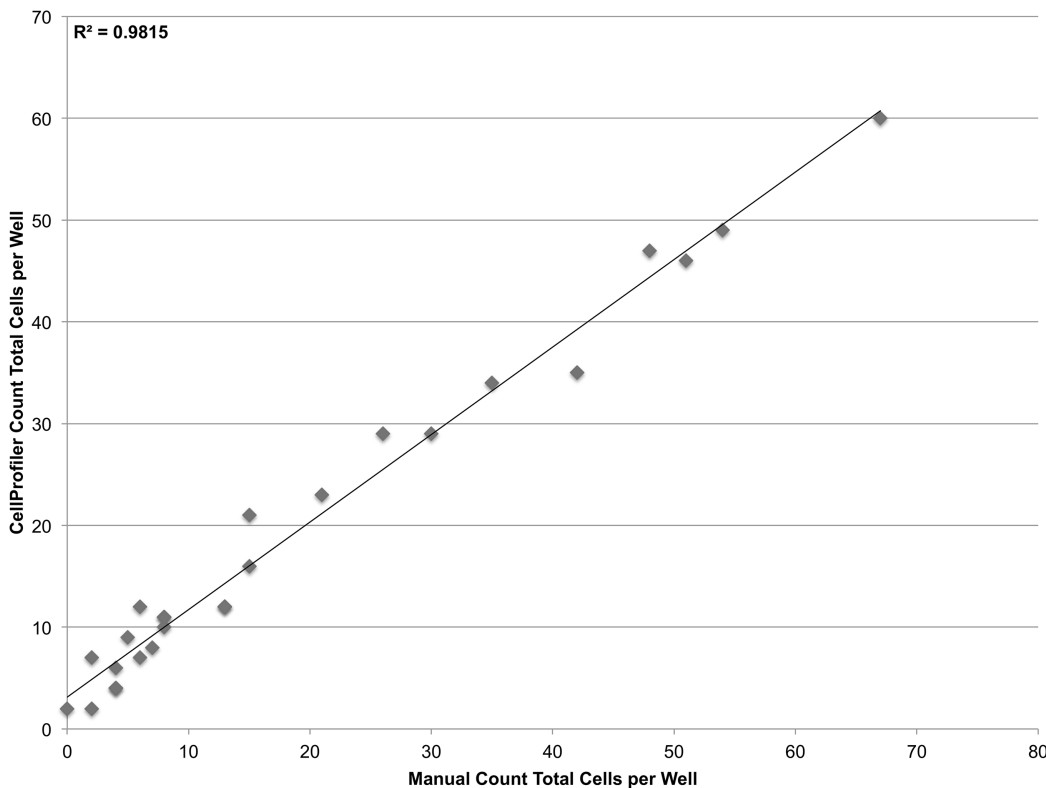

**Figure 5 Establishing sensitivity.** Various concentrations of JC CRL 2116 cells ranging from 100 cells/well down to 1 cell/well were plated on a 48-well plate and given 24 h to adhere to the surface. CellProfiler-derived counts of DAPI stained nuclei were compared with cell counts obtained by human assessment of associated brightfield images for each well. The system performs robustly even at the single-cell level ($R^2 = 0.98$, slope = 0.86) with most deviations attributable to human error when manually counting.

limitation, several additional layers of image processing, including mask generation, need to be performed; these steps are elaborated upon in the Discussion section.

To validate the power of this approach, as well as the efficacy of cytoplasmic and surface staining, a linear dilution series of J774.A1 cells, a mouse macrophage cell line, was plated going from 10,000 cells/well down to 1,000 cells/well. These cells were stained with Vybrant CFDA SE (cytoplasmic stain) prior to plating, and PE-conjugated anti-CD11b antibodies (surface stain) as well as DAPI (nuclear stain) immediately prior to imaging. The system performs less than ideally when segmenting the cytoplasmic or surface stains alone (Fig. S1). When nuclei are used as seeds however, the microscopy-based cytometer performs robustly both for cytoplasmic and surface staining (Fig. 6).

## Quantification of multiple adherent cell types

To demonstrate the ability of the microscopy-based cytometer to determine absolute counts of multiple cell types in a single well, we recreated a macrophage-tumor co-culture immunology experiment that was conducted in 1991 by Novotney et al., as a proof-of-principle test. JC CRL 2116 tumor cells were labeled with Vybrant CFDA SE in their cell culture flask 24 h prior to plating. J774.A1 macrophages were plated first and were either left

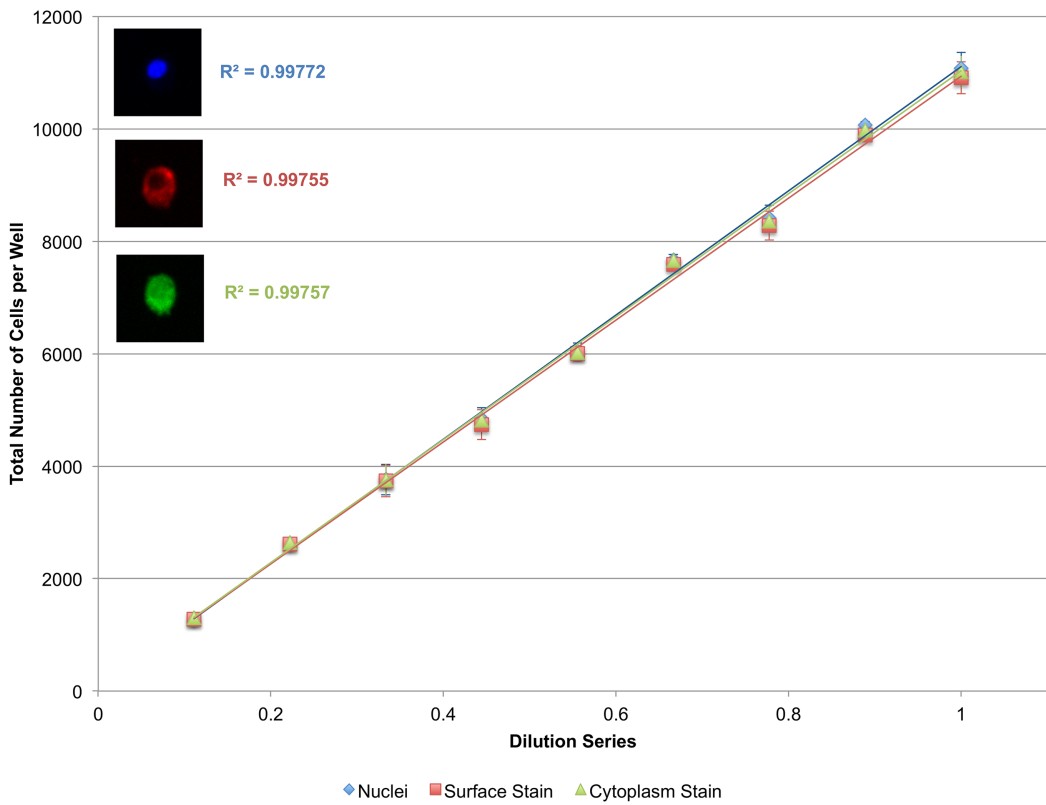

**Figure 6 Secondary quantification of surface and cytoplasmic stains using nuclei as seeds.** A linear dilution series of J774.A1 cells going from 10,000 cells/well down to 1,000 cells/well was plated on a 48-well plate and given 24 h to adhere to the surface. Cells were stained with Vybrant CFDA SE (cytoplasmic stain), phycoerythrin (PE)-conjugated anti-CD11b antibodies (surface stain), and DAPI (nuclear stain). Using nuclei as seeds, cells delineated by a surface or cytoplasmic stain could be accurately quantified as demonstrated by the near-perfect $R^2$. Error bars represent the standard deviation between triplicate conditions.

unprimed or primed with either LPS, IFNγ—a known macrophage activator—or both (*Schoenborn & Wilson, 2007*). After a 24-h incubation, LPS and IFNγ were removed from the wells and the previously stained JC CRL 2216 cells were added. After another 24-h incubation, the wells were stained with a PE-conjugated anti-CD11b targeting antibody to specifically label the J774.A1 macrophages, as well as DAPI to stain all nuclei.

The results of the test reveal an interesting relationship between macrophage activation and tumoricidal activity. The more strongly primed the macrophages (LPS + IFNγ vs unprimed), the more pronounced the tumoricidal response ($P < 0.05$); however, strongly primed macrophages also lose the ability to proliferate, indicated by a sharp drop in macrophage count ($P < 0.05$) (Fig. 7A). Absolute counts of both cell populations show that erroneous results would have been obtained by any setup that looks at relative counts (Fig. 7B). Cytometric analysis reveals that primed J774.A1 macrophages are also larger in size ($P < 0.05$) (Fig. 7C), and appear to have either increased uptake or phagocytic activity, as demonstrated by the retention of Vybrant dye that was initially present in the cytoplasm of the JC CRL 2116 tumor cells ($P < 0.05$) (Fig. 7D). CD11b expression levels do not appear to increase under these experimental conditions ($P > 0.05$) (*Biswas & Sodhi, 2002*).

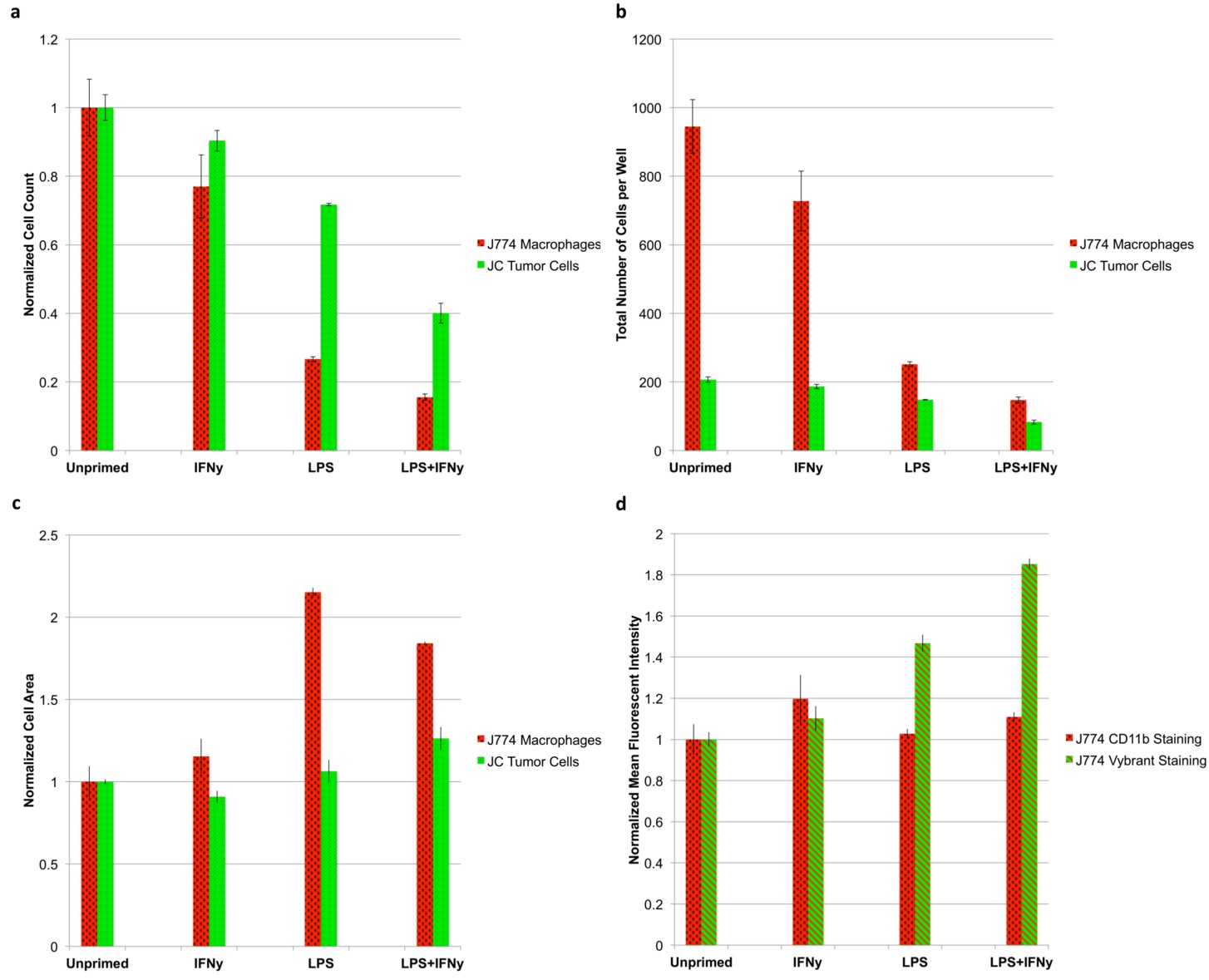

**Figure 7 Tumoricidal properties of primed macrophages in the presence of co-cultured tumor cells.** J774.A1 macrophages were primed with either IFN$\gamma$, LPS, or both and subsequently co-cultured with JC CRL 2116 tumor cells. J774.A1 cells were labeled with PE-conjugated anti-CD11b antibodies (red surface stain) while JC CRL 2116 cells were labeled with Vybrant (green cytoplasmic stain). Both cells were also stained with DAPI. (A) Normalized and (B) absolute cell counts as well as (C) cell area were determined for each condition. (D) CD11b expression levels as well as macrophage uptake/phagocytic activity were calculated by extracting the mean fluorescent intensity of macrophage PE and Vybrant staining, respectively. The experiment reveals an interesting, dynamic relationship between activated macrophages and target tumor cells that would have been missed by any setup that looks at relative counts. Data for (A), (C), and (D) are normalized to the unprimed control. Error bars represent the standard deviation between triplicate conditions.

## DISCUSSION

We present here a relatively simple optical counting setup that brings together several established techniques—including automated microscopy, cellular staining, and cell segmentation—to quantify every cell in an experimental space, without the need for trypsinization or cell scraping. Using whole-well imaging, we demonstrate that taking

representative or random images of a well to make a claim about cell viability/counts may be misleading given the non-uniform distribution of cells in a well. In addition, by counting each individual cell using nuclear segmentation, we demonstrate the impressive accuracy and resolution of our optical quantification system. In general, counting single events/cells provides a resolution that cannot be obtained with bulk ensemble measurements, and can significantly improve sensitivity (*Chang et al., 2012*; *Rissin et al., 2010*). This approach, in conjunction with whole-well imaging, also offers increased statistical power as the experimentally relevant region is sampled in its entirety.

Next, we show that the system can accurately quantify cells within the confluency limit of the plate. By increasing the surface area of each well, however, it is possible to expand the working range of the microscopy-based cytometer. Though 12-well plates were used here to accurately quantify up to 100,000 cells/well, it is possible to use 6-well plates or less, with a larger corresponding surface area per well, to push the upper limits of detection.

On the lower end of the microscopy-based cytometer's operating range, illumination correction, and sufficient post-stain washing make counting of just a single-cell theoretically possible with minimal interference from background or noise. Data from tests that were conducted with only a few cells per well demonstrate the single-cell sensitivity of the system, but slight deviations in segmented nuclei-derived counts versus manual counts are evident as demonstrated by the imperfect $R^2$ and non-unity slope. While manual counts were performed solely on brightfield images, merging of nuclei in the DAPI channel with cell outlines in the brightfield channel reveal that deviations in the lower range data were actually due to human error. Ultimately, counts obtained via CellProfiler proved more reliable than manual counting validating the ability of the microscopy-based cytometer to quantify down to the single-cell level. Systems with improved sensitivity are not only advantageous when conducting screening assays on smaller sized plates (such as 96-, 384-, and 1,536-well plates), but also when detecting a very low concentration of cells on large well plates. A sensitive system may also prove useful for detecting rare cell subpopulations in a heterogeneous group (*Lin et al., 2011*).

Testing of the upper and lower extremes of the microscopy-based cytometer reveal the greater than 5-log detection range that can be achieved with the instrument. Systems with a large dynamic range are particularly advantageous for screening cytotoxic compounds such as antineoplastic agents. It has been shown that compounds that reduce cell viability by at least two orders of magnitude in vitro are more likely to demonstrate a response in clinical trials, making an assay with an even larger dynamic range more appealing (*Frgala et al., 2007*). All together, the combination of improved sensitivity, resolution, and dynamic range afforded by the automated microscopy-based cytometer opens up the possibility of a new set of cell culture experiments that were not previously feasible.

We next show that certain cytoplasmic and surface stains, in conjunction with a nuclear stain, can be used to effectively differentiate cell populations in a co-culture setup. Vybrant CFDA SE dyes, for example, are designed to form intracellular fluorescent conjugates that homogenously stain the cell cytoplasm, are well-retained, and passed onto daughter

cells during division but not transferred to adjacent cells (*Bronner-Fraser, 1985*; *Hodgkin, Lee & Lyons, 1996*; *Lyons & Parish, 1994*; *Nose & Takeichi, 1986*; *Weston & Parish, 1990*). Alternatively, cells can be stained post-experiment and just prior to imaging using target-specific dyes, such as fluorescently-conjugated antibodies.

For counting of more than one cell type in a co-culture setup, using a mask generated from the fluorescent outlines of either a surface or cytoplasmic stain to delineate which nuclei belong to which cell type proved to be the most robust. To do this, whole-cell fluorescence—generated from either a surface or cytoplasmic stain—is used to create an inclusive mask that retains nuclei contained within. This method can be repeated iteratively for each cell specific stain, eventually grouping every nucleus with its associated cell population. These filtered nuclei images can then be segmented and quantified using standard nuclear segmentation to generate respective counts for each cell type. This approach is made possible using the fluorescent staining and algorithmic combination proposed herein, and overcomes many of the limitations of traditional segmenting systems. A sample workflow of this process with associated images can be found in the supplementary section (Fig. S2).

It should be mentioned that many commercial image cytometers largely rely on primary segmentation for the quantification of multiple adherent cell types in a single well (*Pozarowski, Holden & Darzynkiewicz, 2012*). This often requires the utilization of an array of algorithmic functions such as thresholding, contouring, water shedding, cleaning, eroding, dilating, opening, closing, and smoothing to uniquely identify individual cells. Classifiers (both object and pixel) are also occasionally utilized to train systems to recognize user-delineated objects. However, these algorithmic approaches are largely inadequate for quantifying cells with complex morphologies ultimately limiting the accuracy and performance of commercial systems in these settings. For our study in particular, primary segmentation of the cytoplasmic and surface stains alone performed reasonably well when cell morphology was round and staining was homogenous (as was the case with the J774.A1 triple stain). When cell morphology became more spindly (as seen with JC CRL 2116 cells), or staining became more inhomogeneous (as seen with activated J774.A1 cells) however, a marked decrease in primary segmentation performance was observed (data not shown) indicating that the aforementioned algorithmic functions are limited in their generalizability. Using the fluorescent outline of the cell to generate a mask removes the need for primary segmentation of these stains, and instead harnesses the accuracy/power of nuclear segmentation, allowing for highly accurate quantification of multiple cell types with varying morphologies in a co-culture setup. If a user was already in possession of a commercial optical counting system, the proposed algorithmic approach could be adapted to achieve the same results.

The ability of the microscopy-based cytometer to accurately count individual cells of a specific population is best demonstrated by the macrophage-tumor co-culture experiment we recreated. In 1991, *Novotney et al. (1991)* set out to determine the tumoricidal properties of J774.A1 mouse macrophages when primed with LPS. To do this, the target tumor cell line needed to be radiolabeled with $^{51}$Cr then co-cultured with primed J774.A1 macrophages. The extent of tumor killing was determined by measuring

the increase in radioactivity of the supernatant due to tumor cell death and detachment (*Novotney et al., 1991*). Unfortunately, the use of $^{51}$Cr to radiolabel target tumor cells is expensive and manually intensive by today's standards. While cell-permeable fluorogenic protease substrates have been developed as a replacement for $^{51}$Cr, neither is capable of dynamically quantifying changes in effector cell (macrophage) count and the tumor cells they target (*Packard & Komoriya, 2008*). A suitable alternative assay to reliably count both tumor cell and macrophage populations does not exist for this particular experiment. Flow cytometry, for example, would not be preferable here because J774.A1 macrophages are not amenable to trypsin treatment, and thus require manual scraping. In addition, standard flow cytometry would struggle to tease out individual counts of each cell type because the data is collected as a relative count to total number of cells gated. As shown in Fig. 7B, the quantification of relative counts rather than absolute counts would have led the user to believe that macrophage priming actually promotes tumor cell growth as opposed to inhibiting it (tumor cells represent 18.0% of the total sample in the unprimed control vs 36.1% of the total sample in the LPS + IFN$\gamma$ primed condition). The microscopy-based cytometer system we have developed, however, can determine absolute counts of both cell types with single-cell accuracy. With the priming of the J774.A1 macrophages, we were able to not only elucidate the tumoricidal activity of the macrophages, but also the inverse nature of macrophage proliferation and cytotoxic potential.

In addition to cell count, the microscopy-based cytometer was able to extract morphological data, such as cell size and mean fluorescence intensity, for every cell. The system is also capable of extracting additional parametric data, such as cell eccentricity, orientation, number of neighbors, first closest object distance, and granularity, extending its capacity beyond simply counting, and into cytometry. The cytometer is also compatible with other fluorescent-based assay stains including annexin V and/or propidium iodide for assessing cell death, CFSE or Bromodeoxyuridine (BrdU) for measuring cell proliferation, and alamarBlue or calcein AM for determining cell viability. For many fields of biology where the interaction between multiple cell types is important, the ability to conduct this type of co-culture experiment may prove invaluable.

Suggestions for improving and expanding the capacities of the microscopy-based cytometer for co-culture as well as mono-culture studies can be found in the supplementary section. They include recommendations for optimizing workflow, handling of an increasing number of cells/parameters in a single experimental setup (multiplexing), management of multinucleated cells, processing of tissue samples, and analysis of poorly adherent cells. An overview of the technical specifications of the microscopy-based cytometer can also be found in the supplementary (Additional File 4: Text S1).

## CONCLUSION

All together, the microscopy-based cytometer is a viable alternative to flow cytometry, and other currently available imaging cytometers, and is available at only a fraction of the cost for labs that already possess a fluorescent microscope. The system is automated and high throughput using tools already available in most labs. Optical cell counting offers

unprecedented sensitivity, going down to the single-cell level, while still boasting an impressive dynamic range limited only by the size of the well plate used. Further, the use of whole-well images allows for quantification across the entire experimental space conferring single-cell accuracy. In addition to cell counts, other features can be gathered from each experiment, including the spatial distribution of cells as well as various morphological analyses. The system is also compatible with other fluorescent-based assay stains. Since the system is put together by the user, it is highly customizable, and allows for direct assessment of assay performance.

The microscopy-based cytometer has the added benefit of assessing adherent cells directly on the plate without needing to bring them into solution. As a result, the system does not require caustic trypsin treatment to resuspend cells in solution, as is the case with flow cytometry. One of the most promising aspects of the system is that multiple cell types can be plated together and absolute counts of each population can be elucidated. For many fields of biology, absolute counts can help tease out interesting relationships between cell populations that might not be discernible using relative counts. After imaging and analysis, population statistics—including cell counts and fluorescence intensity—can be extracted, thresholded, and displayed in histogram or dot-plot form much like the output of a flow cytometer. Overall, we believe the microscopy-based cytometer will improve the quality of cell cytometric studies and open up the possibility of a new class of experiments centered around the ability to assess multiple cell types in a co-culture setup. The impressive sensitivity and dynamic range of the instrument are also strongly compelling.

## ACKNOWLEDGEMENTS

We would like to thank the Miller lab for providing the Nikon Eclipse Ti-E inverted fluorescent microscope and the Drezek lab for editorial assistance.

### Funding

The authors received no funding for this work.

### Competing Interests

The authors declare that they have no competing interests.

### Author Contributions

- Vishwaratn Asthana conceived and designed the experiments, performed the experiments, analyzed the data, prepared figures and/or tables, authored or reviewed drafts of the paper, approved the final draft.
- Yuqi Tang performed the experiments, analyzed the data, authored or reviewed drafts of the paper, approved the final draft.
- Adam Ferguson performed the experiments, analyzed the data, authored or reviewed drafts of the paper, approved the final draft.
- Pallavi Bugga analyzed the data, authored or reviewed drafts of the paper, approved the final draft.
- Anantratn Asthana performed the experiments, authored or reviewed drafts of the paper, approved the final draft.
- Emily R. Evans analyzed the data, authored or reviewed drafts of the paper, approved the final draft.
- Allen L. Chen analyzed the data, contributed reagents/materials/analysis tools, authored or reviewed drafts of the paper, approved the final draft.
- Brett S. Stern performed the experiments, authored or reviewed drafts of the paper, approved the final draft.
- Rebekah A. Drezek conceived and designed the experiments, contributed reagents/materials/analysis tools, authored or reviewed drafts of the paper, approved the final draft.

## Data Availability

Asthana, Vishwaratn (2018): Immune Co-Culture Images JPEGs. figshare. Fileset. https://doi.org/10.6084/m9.figshare.5926453.v1;

Asthana, Vishwaratn (2018): Lower Range Test JPEGs. figshare. Fileset. https://doi.org/10.6084/m9.figshare.5926411.v1;

Asthana, Vishwaratn (2018): Orders of Magnitude JPEGs. figshare. Fileset. https://doi.org/10.6084/m9.figshare.5926408.v1;

Asthana, Vishwaratn (2018): Linear Test DAPI JPEGs. figshare. Fileset. https://doi.org/10.6084/m9.figshare.5926405.v1;

Asthana, Vishwaratn (2018): J774 DAPI-Antibody-Vybrant Run JPEGs. figshare. Fileset. https://doi.org/10.6084/m9.figshare.5926441.v1.

## Supplemental Information

Supplemental information for this article can be found online at http://dx.doi.org/10.7717/peerj.4937#supplemental-information.

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
