# Peer review of "An inexpensive, customizable microscopy system for the automated quantification and characterization of multiple adherent cell types"

_PeerJ, doi:10.7717/peerj.4937_

## Round 0.1 · original submission · Major Revisions

Please address the concerns of both reviewers, including usability issues as brought up by Reviewer 1 and more context as to related existing methods highlighted by Reviewer 2. Do make sure to include a statistical analysis section as well as improve the readability of the manuscript figures.

Reviewer 1 ·

Basic reporting

This paper is well written and easy to read. I have some minor points that I would like to see corrected or addressed.

Minor Points:


-The acronym MTT isn’t defined (line 52).

-Please make the axes label font size larger in Figure 3 and 7

-I’d remove a few of the stronger modifier adjectives (see “extraordinary” line 234, “incredible” line 310, “remarkable” line 384/409/411)

-I’m not sure what you mean by binary counts on line 409, how does that contribute to the system resolution? You might just want to get rid of that sentence.

Experimental design

The overall design of testing the accuracy of the imaging/segmentation based on serial dilutions is sound. Adding in the option to deal with co-culture situations with secondary stains is a logical extension. I see no significant problems with the experimental design.

The research niche the pipeline fills is correctly identified and appears to be a relevant problem in the sub-field of cell counting in difficult cell types.

Validity of the findings

Overall, I think the paper supports the claims of the software’s performance adequately. I have a few clarifying points that should be fixed.

I don’t think Figure S1 accurately demonstrates whether the automated nuclei segmentation is completely working at very high confluency levels. In order to assess the quality of the segmentation, an overlay of the segmented nuclei and the original image would need to be produced. I do agree with the authors that even on this difficult segmentation task that most nuclei are identified. I would change the wording on line 228 or attempt to measure the accuracy of the confluent segmentations with manual counting.

I’m finding Figure 4 to be a bit confusing. Did the dilution series start at 100,000 or 200,000 cells/well? If the dilution started at 100,000, the first reading at Dilution Series value 1 of about 180,000 is far from the expected count of 100,000. I’m also having trouble visualizing how confluent the wells are at these various dilution series, can you add sample DAPI and segmentation images for some of the points along the series?

Can you also report the slope of the best fit line in Figure 5?

Additional comments

Have you tested your pipelines with CellProfiler 3.0.0 on Mac? I was able to load both pipelines, but received this error when loading S1:

Error while loading SaveImages: Unsupported image type: Objects. Use <i>ConvertObjectsToImage</i> to create an image.
Do you want to stop processing?

And this error when loading S2:

Error while loading ClassifyPixels: Could not find the ClassifyPixels module
Do you want to stop processing?

It looks like ClassifyPixels is provided by Ilastik, which is supposed to be provided by CellProfiler, is it not packaged in CellProfiler 3.0.0? Does Ilastik need to be installed as well on Mac? I’m not sure if either of these errors are show-stoppers, but I thought you might want to know about them.

Reviewer 2 ·

Basic reporting

In general, this technique article complies with all standards envisioned for basic reporting of the journal but needs to be supplemented with some more literature data analysis to outline originality of the work. As for the figures, the axes of some graphs are finely labeled, so enlargement is required.

Experimental design

All experiments are well designed and performed in compliance with the journal’s requirements, but some issues may need to be addressed (see general comments) to further consider the validity of the findings.

Validity of the findings

Validity of the findings is supported by series of replicate experiments. The data are generally robust, statistically sound, and controlled.

Additional comments

In the manuscript entitled “A customizable microscopy system for the automated quantification and characterization of multiple adherent cell types: an alternative to flow cytometry” (#24707) the authors propose an inexpensive microscopy-based automatic system for counting cells with chosen parameters in adherent mono- or co-cultures. In general, ideology of the work is well understood, and the results of are quite interesting and well described. However, the manuscript needs some revision including improvements in editing.

The authors reasonably state that trypsinization or scraping of some types of cells is not suitable for flow cytometric measurements since such treatments may alter viability of cells and some other important cell properties. It should also be stated that in suspension some cells can stick each other thus forming aggregates, a serious disadvantage in flow cytometry- or Coulter-based absolute cell counting. Indeed, at first sight the microscopy-based image analysis technique for counting of adherent cells seems superior. However, this technique may not be ideal either particularly if cells attach to the inner surface of walls of the culture plates or multiwell plates. Usually, in small volume cell cultures using standard multiwell plates (≥ 24 wells), ratios of heights of culture media to diameters of wells are considerably higher than those in large volume cell cultures, assuming that more cells can probably attach to the walls. Obviously, without these cells the absolute cell counting per well would not be complete. This issue needs to be addressed. Otherwise, cell density values (cells/mm2) should be used instead of absolute cell counts to monitor effects.

It is unclear whether similar techniques have been applied by other researchers if standard fluorescent microscope, motorized x-y-z stage, and image analysis software are only needed for cell counting. It should be stated in Introduction. Fluorescence image analysis technique designed for cell quantification in co-cultures was proposed a while ago by Krtolica A. et al. Cytometry 49:73-82 (2002). This work should be referred and compared with the current work in terms of delineating the novelty and possible advantages.

By the way, there are multi-color laser scanning cytometers (Pozarowski P. et al. Methods Mol Biol 931:187-212 (2013)) equipped with a computer controlled stage and autofocus. Although these commercial instruments are not cheap, are they principally different from the current one in terms of image collection, processing and analysis? Notably, in addition to cell quantification, laser scanning cytometry (LSC) like flow cytometry is capable of acquiring cell information sufficient to generate histograms (e.g., DNA content histograms, etc.). More demonstrative and illustrative information on LSC can be found in http://research.uthscsa.edu/facs/PDF%20Files/BCI_Webinar_Telford_2010.pdf

There is also another technique published by Spink B.C. et al. Cell Biol Intl 30:227-238 (2006). They proposed to measure cell numbers in black 96-well microtiter plates with flat and clear bottoms using a fluorescence microplate analyzer. This work should be taken into account and referred as well.

Line 72-73: Intercellular physical contact was shown to be also important for the induction of proliferative responses of bystander cells co-cultured with irradiated cells (Gerashchenko B.I. and Howell R.W. Cytometry Part A 56A:71-80 (2003) can be added to the references).

The term “optical cell cytometer” throughout the text is suggested to be replaced with the term like “microscopy-based cytometer” since in the former term (a) cytometry already implies cell measurements; (b) flow cytometry is an optical technique as well.

In the title of the article the phrase “… an alternative to flow cytometry” is not necessarily.
Can the title sound like “An inexpensive customizable microscopy system for the automated quantification and characterization of multiple adherent cell types”?

The judgment of killing of tumor cells with 51Cr together with intention to recreate this experiment using fluorescent tracers should be moved from Results to Discussion.

How much time does it take to obtain the final results from a single well of 24- or 48-well plate since starting collection of images?
* * *
Specific comments:

The word “unique” or “uniquely” is often used throughout the text.

Line 50: In the phrase “unique limitations” the word “unique” can probably be replaced by the word “apparent” or “obvious”.

Line 51: Can “metabolic” be replaced with “physiologic”?

Lines 52-53: Abbreviation MTT should be uncovered. This assay along with alamarBlue assay should be briefly described.

In Fig. 1, 6-well plates are demonstrated, but experiments were conducted in 24- and 48-well plates.

In the legend of Fig. 2 it is unclear what the type of cells is used.

Fig. 3: Is the average number of cells per image estimated in 24-h cultures? If so, it should be mentioned in the legend.

Fig. 4: Is the total number of cells per is estimated in 24-h cultures? If so, it should be mentioned in the legend.

Lines 286-288: This statement should be placed in Discussion.

Statistical analysis section should be added to Materials and Methods. Do deviations represent SD or SEM of replicate experiments? In Fig. 7, indication of statistical significance (where appropriate) may be helpful to dissect the effect.

---

## Round 0.2 · Minor Revisions

Thank you for addressing the reviewer questions and concerns. A couple of minor points remain. Reviewer 1 mentions that there is still some ambiguity regarding the version of CellProfiler and it seems that addressing this would make it more likely to be pursued by users. In addition, the manuscript does specifically highlight your method's ability to perform absolute cell counts, though Reviewer 2 brings up potential concerns as to whether this is true and/or to what degree. Please try to address this concern, either through the approach suggested by the reviewer or otherwise providing a specific answer to this concern.

Reviewer 1 ·

Basic reporting

The paper is well written and the minor language modifications have improved the text. I see no problems with in this area.

Experimental design

I see no major problems remaining with the experimental design.

Validity of the findings

My previous concerns have been addressed, I see no problems in this area.

Additional comments

I think you should add a short note about the analysis pipeline requiring CellProfiler 2.2.0. I apologize if I missed this addition in your revisions, but a quick search didn’t yield any changes in this respect. I see the largest impact of your paper coming from people looking to use the provided analysis pipeline, but current users of CellProfiler are likely to try to load your pipeline in CellProfiler 3.0.0, see the same errors I did and move on to some other method. A section before “Explanation of CellProfiler Codes and Associated Functions” should fulfill this need.

Reviewer 2 ·

Basic reporting

In general, this technique article complies with all standards envisioned for basic reporting of the journal and now it is supplemented with some more additional literature data. As for the figures, the plots in Fig. 3 are too small, so some enlargement is required.

Experimental design

All experiments are well designed and performed in compliance with the journal’s requirements, however, an issue may need to be addressed (see general comments) to further consider the validity of the findings.

Validity of the findings

Validity of the findings is supported by series of replicate experiments. The data are generally robust, statistically sound, and controlled. The conclusions are appropriately stated and connected to the original question investigated. Also, they are limited to those supported by the results.

Additional comments

There are some improvements in the revised manuscript. However, it still unclear whether this technique is capable of performing precise absolute cell counts per well. In previous commentaries, the possibility was mentioned that some cells may attach to the inner surface of the well’s wall, which is perpendicularly positioned to the well’s bottom (this is not about well’s edge). If they do attach, without them the absolute cell counting per well would not be complete, however. How to clarify this situation? I would probably suggest conducting a simple experiment as follows. Plate an equal numbers of cells into 2 separate wells and culture them for 24 h. Then quantify cells in the entire well by microscopy-based cytometry and compare cell counts with those of another well in which cells were trypsinized and collected for hemocytometry (an old and quite reliable technique). I hope that there will not be any significant differences in cell counts, but this has to be proven. The results of this experiment can be presented as a supplementary material.

---

## Round 0.3 · accepted · Accept

Thank you for addressing the reviewer concerns - congratulations again.

#